# Pharmacokinetics of a single inhalation of hydrogen gas in pigs

**Motoaki Sano** [1,2]*, **Genki Ichihara**[1,2], **Yoshinori Katsumata**[1,2], **Takahiro Hiraide**[1], **Akeo Hirai**[1], **Mizuki Momoi**[1], **Tomoyoshi Tamura** [2,3], **Shigeo Ohata**[5], **Eiji Kobayashi**[1,2,4]

**1** Department of Cardiology, Keio University School of Medicine, Shinjuku-ku, Tokyo, Japan, **2** Center for Molecular Hydrogen Medicine, Keio University, Minato-ku, Tokyo, Japan, **3** Department of Emergency and Critical Care Medicine, Keio University, Shinjuku-ku, Tokyo, Japan, **4** Department of Organ Fabrication, Keio University School of Medicine, Shinjuku-ku, Tokyo, Japan, **5** Department of Neurology, Juntendo University Graduate School of Medicine, Bunkyo-ku, Tokyo, Japan

* msano@a8.keio.jp

**Data Availability Statement:** All relevant data are within the manuscript.

**Funding:** This work was supported by grants from SESA Corporation (E.K.) Sou Hashimoto (Doctors Man Co., Ltd.) provided us with the H2 filling device. The funders had no role in study design,

## Abstract

The benefits of inhaling hydrogen gas ($H_2$) have been widely reported but its pharmacokinetics have not yet been sufficiently analyzed. We developed a new experimental system in pigs to closely evaluate the process by which $H_2$ is absorbed in the lungs, enters the bloodstream, and is distributed, metabolized, and excreted. We inserted and secured catheters into the carotid artery (CA), portal vein (PV), and supra-hepatic inferior vena cava (IVC) to allow repeated blood sampling and performed bilateral thoracotomy to collapse the lungs. Then, using a hydrogen-absorbing alloy canister, we filled the lungs to the maximum inspiratory level with 100% $H_2$. The pig was maintained for 30 seconds without resuming breathing, as if they were holding their breath. We collected blood from the three intravascular catheters after 0, 3, 10, 30, and 60 minutes and measured $H_2$ concentration by gas chromatography. $H_2$ concentration in the CA peaked immediately after breath holding; 3 min later, it dropped to 1/40 of the peak value. Peak $H_2$ concentrations in the PV and IVC were 40% and 14% of that in the CA, respectively. However, $H_2$ concentration decay in the PV and IVC (half-life: 310 s and 350 s, respectively) was slower than in the CA (half-life: 92 s). At 10 min, $H_2$ concentration was significantly higher in venous blood than in arterial blood. At 60 min, $H_2$ was detected in the portal blood at a concentration of 6.9–53 nL/mL higher than at steady state, and in the SVC 14–29 nL/mL higher than at steady state. In contrast, $H_2$ concentration in the CA decreased to steady state levels. This is the first report showing that inhaled $H_2$ is transported to the whole body by advection diffusion and metabolized dynamically.

## Introduction

Inhalation of $H_2$ is reported to have beneficial effects in living organisms [1, 2], and clinical trials have demonstrated its efficacy and safety in patients with acute myocardial infarction [3] and post-resuscitation cardiac arrest [4, 5]. On March 3, 2020, the Chinese National Health and Medical Commission recommended "conditional treatment with hydrogen and oxygen inhalation" in addition to the general oxygen therapy measures in the treatment section of the

data collection and analysis, decision to publish, or preparation of the manuscript.

**Competing interests:** The authors would like to declare the following patents/patent applications associated with this research: S.O. is a founder and CEO of Mitos, Co., Ltd, and the holder of a patent for the use of H2. M.S. and E.K. receive advisory fees from Doctors Man Co., Ltd. M.S. receives advisory fees and research fees from Taiyo Nippon Sanso. Author M.S. is the registered inventor of the following patent jointly filed by Keio University and Taiyo Nippon Sanso. Hydrogen mixed gas supply device for medical purposes (Patent number: 5631524), Medicinal composition for improving prognosis after restart of patient's own heartbeat (Application number PCT/JP2016/088172), Medicinal composition for improving and/or stabilizing circulatory dynamics after onset of hemorrhagic shock (Application number PCT/JP2017/026431). In addition to these, there are three other patents in which the name of the invention is only in Japanese and is not described in English. Here is the name of the invention, which is a literal translation of Japanese into English. Pharmaceutical compositions for reducing weight loss after organ harvesting (Joint application with Keio University and Taiyo Nippon Sanso) Method for generating organ preservation solution containing hydrogen and organ preservation solution containing hydrogen (Joint application with Keio University and Doctors Man; Application number PCT/JP2019/045790). Information about Joint Application between Keio University and Taiyo Nippon Sanso has been uploaded as an attachment. This does not alter our adherence to PLOS ONE policies on sharing data and materials.

Diagnosis and Treatment Protocol for Novel Coronavirus Pneumonia (Trial Version 7), in accordance with a recommendation notification by the Chinese Non-government Medical Institutions Association [6]. However, the kinetics of inhaled $H_2$ in the body have not been sufficiently analyzed to date.

We previously measured, in rats, the time course of $H_2$ levels in different tissues after continuous $H_2$ inhalation, by inserting a needle-type sensor electrode directly into the tissues [7, 8]. However, since the response of the needle-type hydrogen sensor electrode is slow, this makes it unsuitable for measuring short-term changes in $H_2$ concentration in tissues.

In a non-clinical pharmacokinetic study, the distribution of a test drug to various organs and tissues after a single or repeated dose and its change over time should be investigated. In the case of gas, unlike oral and injectable drugs, a non-clinical pharmacokinetic study with a single dose has not been performed. This was because there was no animal protocol for a single-dose study of the gas. The same is true for $H_2$. It remained undetermine whether $H_2$ diffused from the lungs in a blood flow-independent manner or whether $H_2$ was transported throughout the body in a blood flow-dependent manner. Therefore, in the present study, we devised an animal protocol for single-dose inhalation of gas and proved the latter to be true for the first time.

The most effective way of taking $H_2$ into circulating blood after a single inhalation is by fully exhaling, then inhaling 100% $H_2$ to the maximum inspiration position, and holding your breath for as long as you can endure. In the present study, we examine the pharmacokinetics of $H_2$ by replicating this single inhalation method in pigs.

## Materials and methods

### Animals

The present study was designed according to the principles of the ARRIVE (Animal Research: Reporting of In Vivo Experiments) guidelines [9]. Experiments were performed in accordance with the institutional guidelines and the Japanese law on the protection and management of animals. The full ethical proposal was approved by the Research Council and Animal Care and Use Committee of Keio University [approval no: 12094-(7)].

Two female pigs, weighing 22.4 kg and 22.0 kg, were housed in separate cages under temperature- and light-controlled conditions (12-h light/dark cycle) and provided with food and water ad libitum. The pigs were fasted for 12 h prior to surgery, with free access to water. Sedation with medetomidine (0.02 mg/kg) and midazolam (0.1 mg/kg) was followed by endotracheal intubation and mechanical ventilation. Anesthesia was maintained with inhalational isoflurane. Surgery was performed by a surgeon with experience of more than 200 clinical transplant operations, who is a steering member of the transplantation society and a permanent director of the transplantation society of Japan (E.K.).

### Catheter insertion

Before insertion, a catheter (Argyle Medecut LCV-UK kit, 16 GX, 70 cm) was filled with heparinized saline, and the blood collection site was equipped with a three-way stopcock (TERUMO terfusion three-way stopcock, R type). Once at a sufficient depth of anesthesia, the pig was placed in the supine position. A vertical incision of about 10 cm was made in the right side of the neck to expose about 3 cm of the right external jugular vein and the right internal carotid artery (CA). The peripheral side of the right internal CA was ligated with a 1–0 silk thread, a bulldog clip was applied to the medial side, and after An incision was made in CA, a catheter was advanced through the artery to about 5 cm and secured. Subsequently, another prepared catheter was inserted through the right external jugular vein and advanced approximately 25 cm toward the

upper hepatic inferior vena cava and fixed. After confirming sufficient reflux blood could be obtained from both catheters, the skin incision was continuously sutured with 5–0 nylon thread.

Next, a midline incision of the upper abdomen was made about 30 cm below the xiphoid process, and the abdomen was opened. The intestinal tract was held to the left to expose the hepatic portal region. An incision was made in the pancreatic vein and a catheter was inserted about 3 cm toward the hepatic portal. The midline incision was closed by continuous suture with 5–0 nylon thread.

### Protocol for achieving a single inhalation of $H_2$

A median sternum incision was made from the xiphoid process toward the head. At the end of an expiration, the ventilator was removed to stop the animal breathing, and both lungs were manually compressed to mimic maximal forced exhalation by eliminating residual air.

A beach ball was filled with 100% $H_2$ using a hydrogen gas filling device from DAYS (Doctorsman Co., Ltd.) containing hydrogen-absorbing alloys [DAYS (Doctorsman Co., Ltd.)] [10] (Fig 1). The device contained a coupler consisting of a plug and socket with a built-in valve, so when the plug and socket were separated, the inflow of air into the beach ball was completely blocked. The beach ball was connected to a tracheal tube then squeezed, using both hands, at a pressure of about 20 mmHg, to fill the lungs with 100% $H_2$. This reflected the manner in which a bag valve mask would be pressed. The $H_2$-filled lungs were kept at maximal inspiration for approximately 30 s before the tracheal tube was connected to the ventilator and breathing was resumed.

### Blood sampling for $H_2$ concentration measurement

Blood was collected from the three intravascular catheters. First, blood was collected in the steady-state condition (before the breath hold, with the chest open). Next, two sets of experiments were performed per pig. In the first set, blood was collected immediately after the breath

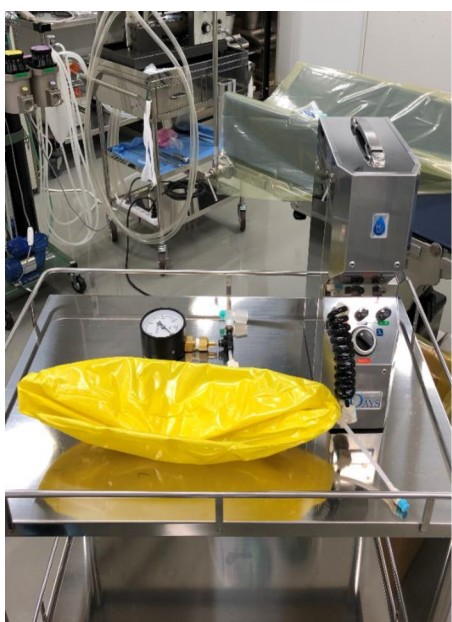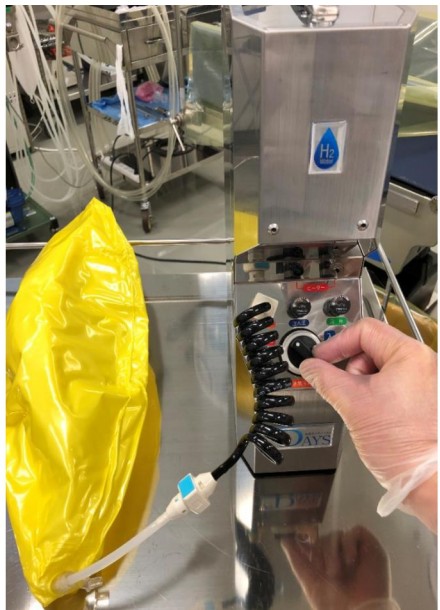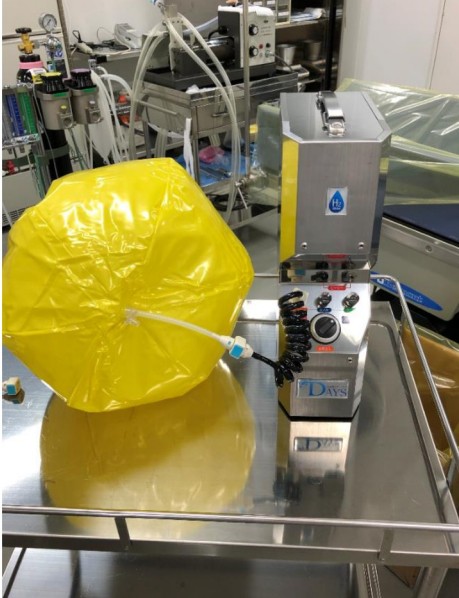

**Fig 1. Beach ball being filled with 100% $H_2$.** The ball is expanding with $H_2$ released from a hydrogen-absorbing alloy in an $H_2$ filling device.

hold and at 3, 10, 30, and 60 min after restarting ventilation. In the second set, blood was collected immediately after the breath hold and at 3 and 10 min after restarting ventilation.

## Measurement of $H_2$ concentration

A needle was inserted into the rubber lid of a 13.5-mL sealed vial, 1 mL of air was extracted, and 1 mL of blood was injected. To prevent outgassing, wax was immediately applied to the rubber lid and the injected hole was sealed.

$H_2$ in the blood was released into the air phase in the closed vial. Some of the air phase (0.2 mL, 0.4 mL or 1 mL, depending on the $H_2$ concentration) was collected from the vial and $H_2$ concentration was measured by gas chromatography (TRIlyzer mBA-3000, Taiyo, Co., Ltd. Osaka, Japan). A calibration curve was obtained using standard $H_2$ gas of 0, 5, 50 and 130 ppm. Each sample was measured twice. The concentration of the sample taken before $H_2$ inhalation was subtracted as background.

## Statistical analysis

Data are expressed as the mean ± standard error of the mean. One-way analysis of variance followed by a Tukey–Kramer multiple comparisons test was used to compare the $H_2$ concentrations between measurement sites. $P < 0.05$ was considered significant. All data were analyzed using GraphPad Prism 8.4 (GraphPad Software Inc., La Jolla, CA).

# Results

## $H_2$ concentration in circulating blood at steady state

Mammalian cells do not produce $H_2$ as they lack the hydrogenase activity necessary for its formation. Instead, resident bacteria in the colon produce a considerable amount of $H_2$ via anaerobic fermentation of unabsorbed carbohydrates. It is generally assumed that $H_2$ produced by bacterial fermentation in the colon is transferred to the portal circulation and excreted through the breath. In a previous breath gas analysis we conducted in healthy volunteers, we found that the concentration of $H_2$ in the breath varies widely (1–56 ppm) between individuals [11].

In the present experiment, we detected minimal $H_2$ in the carotid artery (CA) in the steady-state condition in both pigs. In contrast, $H_2$ concentration in the portal vein (PV) was 67 nL/mL and 8.8 nL/mL for the first and second pigs, respectively, and in the supra-hepatic inferior vena cava (IVC) it was 18 nL/mL and 1.9 nL/mL, respectively. We expect that the large difference between the two pigs in PV $H_2$ concentration is due to differences in $H_2$ production ability by colonic bacteria.

These results indicate that $H_2$ produced by bacteria in the colon is carried by the portal circulation, most of it is trapped in the liver, and the remaining $H_2$ is excreted from the lungs.

## Pharmacokinetics a single inhalation of $H_2$

In the first set of experiments, blood $H_2$ concentration was tracked until 60 min after breathing was resumed. After that, breathing was stopped again at the end of an exhalation and the lungs were manually compressed to reduce residual air, and then expanded with 100% $H_2$ for the second set of experiments. In the second set, $H_2$ concentration in the circulating blood was monitored for 10 min.

Immediately after the end of the breath hold, the peak $H_2$ concentration in the CA of the first pig was 5000 nL/mL in the first set of experiments and 7900 nL/mL in the second set. In the second pig, the concentrations were 10000 and 11000 nL/mL, respectively. (**Fig 2A**). $H_2$ concentration of a 100% aqueous solution was 17,600 nL/mL, meaning that peak $H_2$ in the CA

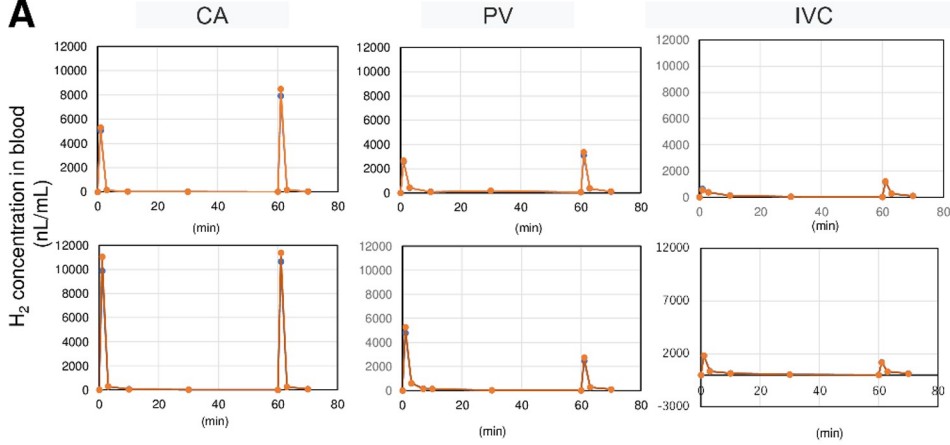

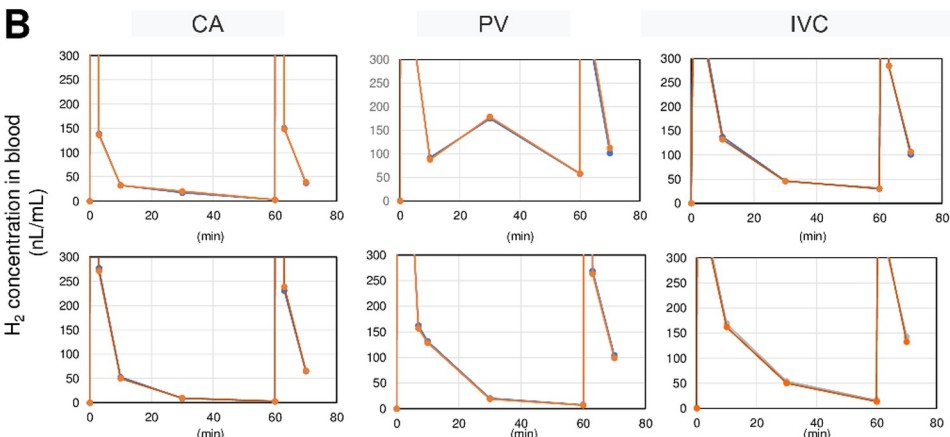

**Fig 2. Time course of blood concentration of H$_2$ after a single inhalation.** (A) Peak H$_2$ concentration in the CA reached 28–60% saturation. Peak H$_2$ concentration in venous blood (PV, IVC) was much lower than that in arterial blood (CA>>PV>IVC). (B) Enlarged low-concentration areas from Fig 3A. After 10 min, blood H$_2$ concentrations were highest in the IVC, then the PV, and lowest in the CA. After 60 min, H$_2$ concentration in the PV and IVC remained higher than at steady state. In A and B, upper and lower graphs in each panel show readings from pig 1 and pig 2, respectively. Duplicate H$_2$ concentration measurements are overlaid. CA, carotid artery; PV, portal vein; IVC, supra-hepatic inferior vena cava.

reached 28–60% saturation by this inhalation method. Peak H$_2$ concentration in venous blood (PV, IVC) was much lower than that in arterial blood (CA>>PV>IVC). This indicates that H$_2$ is not simply diffused, but diffuses while being carried by the bloodstream (advection diffusion), and most H$_2$ is consumed by the tissues.

H$_2$ concentration decreases rapidly in arterial blood (half-life: 92 s) but more slowly in venous blood (half-life: PV, 310 s; IVC, 350 s) (**Fig 2B**). Consequently, H$_2$ concentration after 10 min was greatest in the IVC, then in the PV, and lowest in the CA (**Fig 3**)—the opposite to that at peak concentration. At 60 min after resuming breathing, H$_2$ in the CA had almost disappeared (2.5 nL/mL) (**Fig 4**), but higher concentrations of H$_2$ were detected in the PV and IVC than at steady state. These results indicate that H$_2$ is absorbed in the tissues, then gradually exits the tissues and returns to the heart via venous flow. In other words, a considerable amount of H$_2$ remains in the tissues throughout the body even 60 min after inhalation of H$_2$.

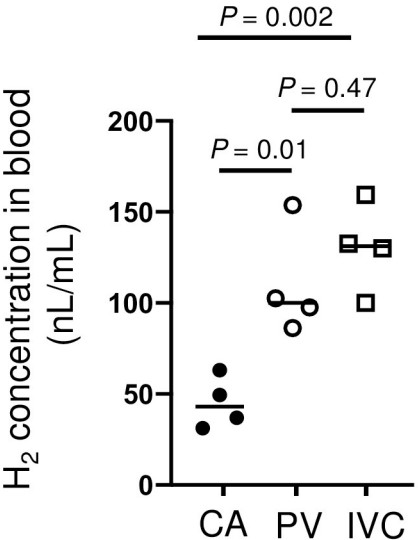

**Fig 3. Blood concentration of H$_2$ 10 min after a single inhalation.** Venous H$_2$ concentrations (PV, IVC) were significantly higher than arterial H$_2$ concentrations (CA). Individual readings (two readings from two animals from each intravascular catheter) and means are shown.

## Discussion

This is the first preclinical study to investigate the kinetics of single-dose inhalation of H$_2$ in the body. We devised a protocol that allows pigs to inhale H$_2$ only once. The ventilator was removed from the intubated pig at the end of expiration. Both lungs were compressed by hand to release the remaining air. We defined this state as the estimated position at maximum exhalation. A beach ball filled with 100% H$_2$ was connected to a tracheal tube and H$_2$ was pumped into the lungs until inflated to the estimated position of maximal inspiration by squeezing the

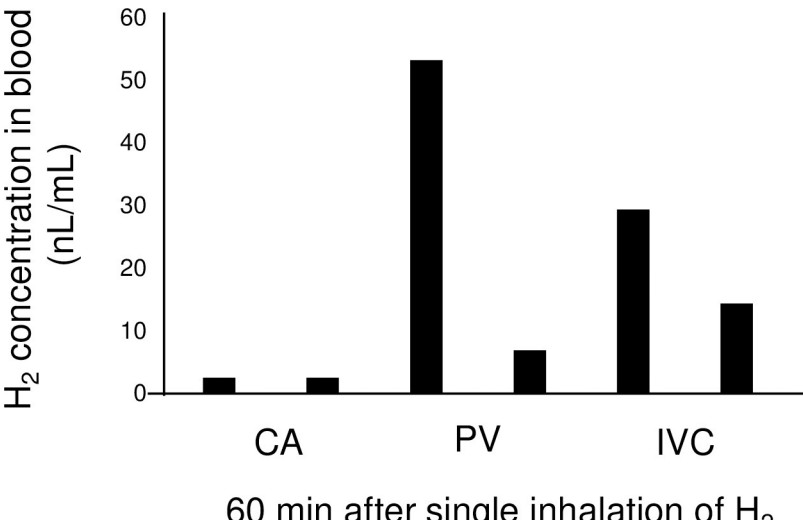

**Fig 4. Blood concentration of H$_2$ 60 min after a single inhalation.** By 60 min, H$_2$ had almost disappeared in the CA but remained in the IVC and PV. Data are one reading per animal from each intravascular catheter.

beach ball with both arms. We kept the $H_2$-inflated lungs intact for a while. We modeled the behavior of holding the breath after inhaling as much $H_2$ as possible using this series of methods.

Since many animal experiments [12–14] and clinical studies [4, 5, 15] have been conducted to examine the protective effect of $H_2$ inhalation on the brain, the CA was chosen as the first blood collection point to prove that the inhaled $H_2$ can reach the brain efficiently. The liver has a dual blood supply from the PV and the hepatic artery. About 75% of the blood flow to the liver comes from the PV and 25% from the hepatic artery. Oxygen is supplied by the PV and the hepatic artery in half each. Therefore, in contrast to the brain, the liver has been regarded as the organ where inhaled $H_2$ is least likely to reach [16]. The liver is the largest organ in the body, performing a number of functions that are essential for life, such as metabolism, detoxification, and excretion, so protecting it with $H_2$ is considered to be a great advantage. We wanted to find out how much $H_2$ is consumed as it passes through the liver, so we compared the $H_2$ concentrations in the PV and supra-hepatic IVC. The $H_2$ concentration of CA immediately after inhalation was very high, and it was confirmed that the inhaled $H_2$ reached the brain efficiently. The peak $H_2$ concentrations of PV and IVC were 40% and 14% of CA, respectively, indicating that inhaled $H_2$ is relatively difficult to reach the liver, but the liver actively consumes the $H_2$.

$H_2$ circulates throughout the body, with only about 10% returning to the venous blood. The arterial blood $H_2$ concentration drops rapidly and has a half-life of about 90 seconds. On the other hand, the half-life of venous blood $H_2$ concentrations is longer, 310 seconds for PV and 350 seconds for IVC; therefore, 3 minutes after inhalation, $H_2$ concentrations in venous blood exceed those in arterial blood. This is presumed to be due to the fact that the $H_2$ diffused into the tissues, which was not metabolized, is gradually returned to the venous blood. $H_2$ can still be detected in venous blood an hour after a single inhalation, but $H_2$ is almost undetectable in arterial blood, perhaps because it is discarded from the lungs.

Gaseous molecules, such as oxygen, carbon dioxide, nitric oxide, and hydrogen sulfide, can bind to the ferrous heme of a variety of proteins with high affinity; thus, they bind to hemoglobin. $H_2$, however, does not bind to heme, and its receptor molecules and their downstream effectors have not yet been identified. Inhaled $H_2$ is simply dissolved in the plasma and transported to the whole body. Supply of $H_2$ via the arterial blood to the tissues depends on blood flow. However, unlike for oxygen, there is no system that keeps $H_2$ concentrated in the blood vessels, so it diffuses out of the blood vessels as it travels.

Whether $H_2$ is inhaled or drunk in water enriched with dissolved $H_2$ [17, 18], breath analysis shows that 60% is excreted in the breath, with 40% being consumed by the body. The amount of $H_2$ released from the body surface is estimated to be extremely small—about 0.1% [17]. By comparing the $H_2$ concentration in the PV and IVC, we estimate that 64% of $H_2$ in the portal blood is trapped just by passing through the liver. Together, these results indicate that $H_2$ is consumed by the body, but the molecular mechanism of how $H_2$ is metabolized remains unknown.

We, at the Center for Molecular Hydrogen Medicine at Keio University in Tokyo, have demonstrated the therapeutic effects of $H_2$ on diseases such as acute myocardial infarction [3, 7], post-cardiac arrest syndrome [4, 5, 13, 14], hemorrhagic shock [19, 20], and organ transplantation [10] in both animal experiments and clinical studies. In patients with severe COVID-19, the immune over-response causes the production of large amounts of cytokines by alveolar macrophages, which becomes a cytokine storm, resulting in the progression of acute respiratory distress syndrome, abnormal blood coagulation, and multiple organ failure [21]. $H_2$ gas not only inhibits the overproduction of cytokines [13], but also suppresses vascular endothelial damage [20], facilitates the flow of red blood cells in microvessels and increases the efficiency of gas exchange (M.S. unpublished observation). Accordingly, $H_2$ inhalation

therapy has great potential to improve the life expectancy of intubated COVID-19 patients admitted to the intensive care unit with severe hypoxemia. The HYBRID II Trial (Efficacy of inhaled HYdrogen on neurological outcome following BRain Ischemia During out-of-hospital cardiac arrest), a multicenter, randomized, double-blind, placebo-controlled, controlled clinical trial investigating the efficacy of $H_2$ inhalation therapy for patients after out-of-hospital cardiac arrest, has been underway since 2017 using a hydrogenated oxygen supply device jointly developed by Keio University and Taiyo Nippon Sanso (jRCTs031180352) [4]. In the HYBRID II trial, patients after cardiopulmonary arrest and resuscitation have been treated with hydrogenated oxygen for 18 hours in combination with conventional cooling methods. Prior to Hybrid II, we had conducted an open-label, single-arm, prospective interventional trial at Keio University Hospital in Tokyo in 2014 to evaluate the feasibility and safety of $H_2$ inhalation in patients with out-of-hospital cardiac arrest achieved a spontaneous return of circulation [5]. Non-cardiogenic cardiac arrest patients were also enrolled in this TRIAL; 5 patients were entered, one of whom was a CA patient due to severe pneumonia and septic shock. This patient had a stable respiratory state during $H_2$ inhalation, but died after $H_2$ inhalation was completed due to a rapid deterioration of the respiratory state. Based on this experience, we are considering a study protocol in which patients with severe COVID-19 disease will continue to receive hydrogenated oxygen for at least one week until their hypoxemia has sufficiently improved. In China, the hydrogen-oxygen mixture gas inhaler was certified as a national class III medical device in February, and it is reported to have a certain effect on the improvement of hypoxic symptoms of COVID-19-related pneumonia. We hope to bring this new treatment to patients as quickly as possible in order to save the lives of severely ill patients with COVID-19.

## Conclusion

We developed a pig model in which we could study the pharmacokinetics of a single inhalation of $H_2$. Inhaled $H_2$ was transported to the whole body by advection diffusion, and metabolized dynamically. The present results will contribute to the knowledge on $H_2$ biology that is increasingly being applied to medicine.

## Acknowledgments

The authors are grateful to Yasuyo Aoyama (Doctors Man Co., Ltd.), Keiji Kawagoe (Toku Corporation), Suga Kato (JHyPA) and Mayumi Takeda (JHyPA) for technical assistance.

## Author Contributions

**Data curation:** Motoaki Sano, Genki Ichihara, Yoshinori Katsumata, Takahiro Hiraide, Akeo Hirai, Mizuki Momoi, Eiji Kobayashi.

**Formal analysis:** Motoaki Sano, Tomoyoshi Tamura, Shigeo Ohata.

**Funding acquisition:** Motoaki Sano.

**Investigation:** Motoaki Sano.

**Methodology:** Motoaki Sano, Eiji Kobayashi.

**Project administration:** Motoaki Sano.

**Supervision:** Motoaki Sano.

**Writing – original draft:** Motoaki Sano.

**Writing – review & editing:** Motoaki Sano.

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
