## [Decision Letter · Decision Letter 0]

5 May 2020

PONE-D-20-11712

Pharmacokinetics of a single inhalation of hydrogen gas in pigs

PLOS ONE

Dear Dr. Sano,

Thank you for submitting your manuscript to PLOS ONE. After careful consideration, we feel that it has merit but does not fully meet PLOS ONE’s publication criteria as it currently stands. Therefore, we invite you to submit a revised version of the manuscript that addresses the points raised during the review process.

We would appreciate receiving your revised manuscript by Jun 19 2020 11:59PM. To enhance the reproducibility of your results, we recommend that if applicable you deposit your laboratory protocols in protocols.io, where a protocol can be assigned its own identifier (DOI) such that it can be cited independently in the future. For instructions see: http://journals.plos.org/plosone/s/submission-guidelines#loc-laboratory-protocols

We look forward to receiving your revised manuscript.

Kind regards,

Tohru Minamino, M.D., Ph.D.

Academic Editor

PLOS ONE

2. We note that you have stated that Animals were handled in accordance with the Animal (Scientific Procedures) Act 1986 of the United Kingdom. However, the study appears to have been conducted in Japan. Please confirm that you abided by all local regulations relating the use of animals in scientific procedures.

"S.O. is a founder and CEO of Mitos, Co., Ltd, and the holder of a patent for the use of H2. M.S. and E.K. receive advisory fees from Doctors Man Co., Ltd. M.S. receives advisory fees and research fees from Taiyo Nippon Sanso."

4. We note that you have a patent relating to material pertinent to this article. Please provide an amended statement of Competing Interests to declare this patent (with details including name and number), along with any other relevant declarations relating to employment, consultancy, patents, products in development or modified products etc. Please confirm that this does not alter your adherence to all PLOS ONE policies on sharing data and materials, as detailed online in our guide for authors http://journals.plos.org/plosone/s/competing-interests by including the following statement: "This does not alter our adherence to  PLOS ONE policies on sharing data and materials.” If there are restrictions on sharing of data and/or materials, please state these. Please note that we cannot proceed with consideration of your article until this information has been declared.

Reviewers' comments:

Reviewer's Responses to Questions

**Comments to the Author**

1. Is the manuscript technically sound, and do the data support the conclusions?

Reviewer #1: Yes

Reviewer #2: No

Reviewer #3: Yes

2. Has the statistical analysis been performed appropriately and rigorously? 

Reviewer #1: Yes

Reviewer #2: No

Reviewer #3: Yes

3. Have the authors made all data underlying the findings in their manuscript fully available?

Reviewer #1: Yes

Reviewer #2: Yes

Reviewer #3: Yes

4. Is the manuscript presented in an intelligible fashion and written in standard English?

Reviewer #1: Yes

Reviewer #2: Yes

Reviewer #3: Yes

5. Review Comments to the Author

Reviewer #1: This study is a non-clinical pharmacokinetic study of hydrogen in pigs. In addition to the reduction of ischemia-reperfusion injury in the emergency room, hydrogen is also expected to be effective in the treatment of pneumonia with COVID-19, which is currently undergoing clinical trials.　Non-clinical pharmacokinetic studies are required to obtain regulatory approval, but no credible experimental results have been reported to date. This study investigates in detail how hydrogen absorbed from the lungs is transported into the body and metabolized. It is a very well-prepared experiment. It is also valuable in terms of non-clinical pharmacokinetic studies in pigs. I'll only mention minor comments.

Comments

1). As COVID-19 spread across the globe, scientists are hunting for drugs that could be repurposed as a quick way to tackle the disease. Hydrogen is one such candidate. Therefore, please describe in the discussion the possible mechanisms by which hydrogen may be effective in the treatment of COVID-19-related pneumonia.

2). Why were blood concentrations monitored at three locations: the carotid artery and portal vein and the supra-hepatic inferior vena cava?

Reviewer #2: To the editor

As a versatile medical gas candidate, hydrogen has attracted much attention in Asia, especially in Japan, China, and Korea, and has been subjected to animal experiments and clinical trials. To date, the only studies to examine the distribution of hydrogen in the body after the hydrogen intake have been conducted in rats, which are small animals. It is significant to investigate the pharmacokinetics of hydrogen in pigs, which are large animals with anatomical similarities to humans. The accuracy of the experiment is high and the results are credible. I would like to offer some minor comments.

To the authors

This study examined how hydrogen absorbed from the lungs diffuses throughout the body when pigs are given a single dose of hydrogen inhalation. After collapsing the lungs by opening the chest, the remaining air in the lungs is expelled by hand compression, and then the lungs are inflated with hydrogen to the level of the maximum inspiratory volume. By maintaining this state for a while, the authors create a state in which the maximum possible dose of hydrogen is taken up from the lungs into the blood. Blood samples are drawn from the carotid artery, portal vein, and supra-hepatic IVC to monitor the kinetics of hydrogen concentrations in the blood. The authors conclude that hydrogen gas is transported to each organ in a blood flow-dependent manner, where it is actively metabolized. As a versatile medical gas candidate, hydrogen has attracted much attention in Asia, especially in Japan, China, and Korea, and has been subjected to animal experiments and clinical trials. To date, the only studies to examine the distribution of hydrogen in the body after the hydrogen intake have been conducted in rats, which are small animals. It is significant to investigate the pharmacokinetics of hydrogen in pigs, which are large animals with anatomical similarities to humans. The accuracy of the experiment is high and the results are credible. I would like to offer some minor comments.

Comments

(1) This paper examines the pharmacokinetics after hydrogen inhalation, but for the general reader, it should touch on the known pharmacology of hydrogen, such as its antioxidant and anti-inflammatory effects.

(2) Hydrogen can be taken into the body not only by inhalation, but also by drinking hydrogen-containing water. What is the pharmacokinetics of hydrogen dissolved in orally ingested water, in analogy with the present experimental results?

Reviewer #3: In this study, the concentration of hydrogen in the blood was measured at three locations: the carotid artery, the portal vein, and the supra-hepatic inferior inferior vena cava, after a single inhalation of hydrogen gas in pigs. Although many preclinical and clinical trials have shown the therapeutic effect of hydrogen on a variety of conditions involving ischemia-reperfusion injury, information on how ingested hydrogen is distributed throughout the body is scarce. It is significant that the present study proved that hydrogen absorbed from the lungs is transported by advection diffusion in the bloodstream to systemic organs, rather than by simple diffusion. This study is presented by a group that is leading the world in non-clinical and clinical research on hydrogen gas. The surgery is performed by a transplant surgeon with extensive experience in pig surgery. The concentration of hydrogen was measured by gas chromatography with little error between the two measurements, and the accuracy of the data is high. This data is necessary for hydrogen to be approved by the pharmaceutical industry as a medical gas in the future. I'll only mention minor comments.

Comment:

(1) Why did you insist on single-dose inhalation instead of continuous inhalation?

(2) The treatment of pneumonia caused by COVID-19 has started in Wuhan, China. Please describe the expected pharmacological effects of hydrogen in this clinical setting.

6. PLOS authors have the option to publish the peer review history of their article (what does this mean?). If published, this will include your full peer review and any attached files.

Reviewer #1: No

Reviewer #2: No

Reviewer #3: No

---

## [Author Response · Author response to Decision Letter 0]

19 May 2020

Comments to the Author

Reviewer #1: 

This study is a non-clinical pharmacokinetic study of hydrogen in pigs. In addition to the reduction of ischemia-reperfusion injury in the emergency room, hydrogen is also expected to be effective in the treatment of pneumonia with COVID-19, which is currently undergoing clinical trials.　Non-clinical pharmacokinetic studies are required to obtain regulatory approval, but no credible experimental results have been reported to date. This study investigates in detail how hydrogen absorbed from the lungs is transported into the body and metabolized. It is a very well-prepared experiment. It is also valuable in terms of non-clinical pharmacokinetic studies in pigs. I'll only mention minor comments.

Comments

1). As COVID-19 spread across the globe, scientists are hunting for drugs that could be repurposed as a quick way to tackle the disease. Hydrogen is one such candidate. Therefore, please describe in the discussion the possible mechanisms by which hydrogen may be effective in the treatment of COVID-19-related pneumonia.

Response

Thank you very much for kind remarks. We, at the Center for Molecular Hydrogen Medicine at Keio University in Tokyo, have demonstrated the therapeutic effects of H2 on diseases such as acute myocardial infarction [3, 7], post-cardiac arrest syndrome [4, 5, 13, 14], hemorrhagic shock [19, 20], and organ transplantation [10] in both animal experiments and clinical studies. In patients with severe COVID-19, the immune over-response causes the production of large amounts of cytokines by alveolar macrophages, which becomes a cytokine storm, resulting in the progression of acute respiratory distress syndrome, abnormal blood coagulation, and multiple organ failure [21]. H2 gas not only inhibits the overproduction of cytokines [13], but also suppresses vascular endothelial damage [20], facilitates the flow of red blood cells in microvessels and increases the efficiency of gas exchange (M.S. unpublished observation). Accordingly, H2 inhalation therapy has great potential to improve the life expectancy of intubated COVID-19 patients admitted to the intensive care unit with severe hypoxemia. The HYBRID II Trial (Efficacy of inhaled HYdrogen on neurological outcome following BRain Ischemia During out-of-hospital cardiac arrest), a multicenter, randomized, double-blind, placebo-controlled, controlled clinical trial investigating the efficacy of H2 inhalation therapy for patients after out-of-hospital cardiac arrest, has been underway since 2017 using a hydrogenated oxygen supply device jointly developed by Keio University and Taiyo Nippon Sanso (jRCTs031180352)[4]. In the HYBRID II trial, patients after cardiopulmonary arrest and resuscitation have been treated with hydrogenated oxygen for 18 hours in combination with conventional cooling methods. Prior to Hybrid II, we had conducted an open-label, single-arm, prospective interventional trial at Keio University Hospital in Tokyo in 2014 to evaluate the feasibility and safety of H2 inhalation in patients with out-of-hospital cardiac arrest achieved a spontaneous return of circulation [5]. Non-cardiogenic cardiac arrest patients were also enrolled in this TRIAL; 5 patients were entered, one of whom was a CA patient due to severe pneumonia and septic shock. This patient had a stable respiratory state during H2 inhalation, but died after H2 inhalation was completed due to a rapid deterioration of the respiratory state. Based on this experience, we are considering a study protocol in which patients with severe COVID-19 disease will continue to receive hydrogenated oxygen for at least one week until their hypoxemia has sufficiently improved. In China, the hydrogen-oxygen mixture gas inhaler was certified as a national class III medical device in February, and it is reported to have a certain effect on the improvement of hypoxic symptoms of COVID-19-related pneumonia. We hope to bring this new treatment to patients as quickly as possible in order to save the lives of severely ill patients with COVID-19.

2). Why were blood concentrations monitored at three locations: the carotid artery and portal vein and the supra-hepatic inferior vena cava?

In the Discussion section, I added the reasons why I measured the hydrogen concentration at the three locations as follows.

As several clinical studies have been reported or are underway to validate the efficacy of hydrogen inhalation for brain damage, the carotid artery was chosen as the first blood collection point to prove that the inhaled hydrogen can reach the brain efficiently.

The liver has a dual blood supply from the portal vein and the hepatic artery. About 75% of the blood flow to the liver comes from the portal vein and 25% from the hepatic artery. Oxygen is supplied by the portal vein and the hepatic artery in half each. Therefore, in contrast to the brain, the liver has been regarded as the organ where inhaled hydrogen is least likely to reach. The liver is the largest organ in the body, performing a number of functions that are essential for life, such as metabolism, detoxification, and excretion, so protecting it with H2 is considered to be a great advantage. Therefore, we wanted to find out how much H2 is consumed as it passes through the liver, so we compared the H2 concentrations in the portal vein and supra-hepatic IVC.

Reviewer #2: 

This study examined how hydrogen absorbed from the lungs diffuses throughout the body when pigs are given a single dose of hydrogen inhalation. After collapsing the lungs by opening the chest, the remaining air in the lungs is expelled by hand compression, and then the lungs are inflated with hydrogen to the level of the maximum inspiratory volume. By maintaining this state for a while, the authors create a state in which the maximum possible dose of hydrogen is taken up from the lungs into the blood. Blood samples are drawn from the carotid artery, portal vein, and supra-hepatic IVC to monitor the kinetics of hydrogen concentrations in the blood. The authors conclude that hydrogen gas is transported to each organ in a blood flow-dependent manner, where it is actively metabolized. As a versatile medical gas candidate, hydrogen has attracted much attention in Asia, especially in Japan, China, and Korea, and has been subjected to animal experiments and clinical trials. To date, the only studies to examine the distribution of hydrogen in the body after the hydrogen intake have been conducted in rats, which are small animals. It is significant to investigate the pharmacokinetics of hydrogen in pigs, which are large animals with anatomical similarities to humans. The accuracy of the experiment is high and the results are credible. I would like to offer some minor comments.

Comments

(1) This paper examines the pharmacokinetics after hydrogen inhalation, but for the general reader, it should touch on the known pharmacology of hydrogen, such as its antioxidant and anti-inflammatory effects.

Response

Thank you very much for summarizing the manuscript and for your positive comments.

The biological effects of H2 are explained in relation to the expected therapeutic effects of H2 on COVID-19 in the discussion section.

“In patients with severe COVID-19, the immune over-response causes the production of large amounts of cytokines by alveolar macrophages, which becomes a cytokine storm, resulting in the progression of acute respiratory distress syndrome, abnormal blood coagulation, and multiple organ failure [21]. H2 gas not only inhibits the overproduction of cytokines [13], but also suppresses vascular endothelial damage [20], facilitates the flow of red blood cells in microvessels and increases the efficiency of gas exchange (M.S. unpublished observation). Accordingly, H2 inhalation therapy has great potential to improve the life expectancy of intubated COVID-19 patients admitted to the intensive care unit with severe hypoxemia. The HYBRID II Trial (Efficacy of inhaled HYdrogen on neurological outcome following BRain Ischemia During out-of-hospital cardiac arrest), a multicenter, randomized, double-blind, placebo-controlled, controlled clinical trial investigating the efficacy of H2 inhalation therapy for patients after out-of-hospital cardiac arrest, has been underway since 2017 using a hydrogenated oxygen supply device jointly developed by Keio University and Taiyo Nippon Sanso (jRCTs031180352)[4]. In the HYBRID II trial, patients after cardiopulmonary arrest and resuscitation have been treated with hydrogenated oxygen for 18 hours in combination with conventional cooling methods. Prior to Hybrid II, we had conducted an open-label, single-arm, prospective interventional trial at Keio University Hospital in Tokyo in 2014 to evaluate the feasibility and safety of H2 inhalation in patients with out-of-hospital cardiac arrest achieved a spontaneous return of circulation [5]. Non-cardiogenic cardiac arrest patients were also enrolled in this TRIAL; 5 patients were entered, one of whom was a CA patient due to severe pneumonia and septic shock. This patient had a stable respiratory state during H2 inhalation, but died after H2 inhalation was completed due to a rapid deterioration of the respiratory state. Based on this experience, we are considering a study protocol in which patients with severe COVID-19 disease will continue to receive hydrogenated oxygen for at least one week until their hypoxemia has sufficiently improved”. 

(2) Hydrogen can be taken into the body not only by inhalation, but also by drinking hydrogen-containing water. What is the pharmacokinetics of hydrogen dissolved in orally ingested water, in analogy with the present experimental results?

Response

If someone drinks H2-rich water, it will be absorbed from the gastrointestinal tract and will first pass through the liver. Our experimental results suggest that most H2 is actively consumed by the liver. Some H2 passes through the liver, but most of it will be discarded in the atmosphere while passing through the lungs. Certainly, the concentration of H2 in the exhaled air after drinking the H2-rich water increases. How much H2 reaches the left ventricle and is supplied to the whole body can be determined by feeding pigs high H2 water and then measuring the H2 concentration in a similar manner.

Reviewer #3: In this study, the concentration of hydrogen in the blood was measured at three locations: the carotid artery, the portal vein, and the supra-hepatic inferior inferior vena cava, after a single inhalation of hydrogen gas in pigs. Although many preclinical and clinical trials have shown the therapeutic effect of hydrogen on a variety of conditions involving ischemia-reperfusion injury, information on how ingested hydrogen is distributed throughout the body is scarce. It is significant that the present study proved that hydrogen absorbed from the lungs is transported by advection diffusion in the bloodstream to systemic organs, rather than by simple diffusion. This study is presented by a group that is leading the world in non-clinical and clinical research on hydrogen gas. The surgery is performed by a transplant surgeon with extensive experience in pig surgery. The concentration of hydrogen was measured by gas chromatography with little error between the two measurements, and the accuracy of the data is high. This data is necessary for hydrogen to be approved by the pharmaceutical industry as a medical gas in the future. I'll only mention minor comments.

Comment:

(1) Why did you insist on single-dose inhalation instead of continuous inhalation?

Response

Thank you very much for summarizing the manuscript and for your positive comments. An explanation of why a single inhalation test was performed has been added to the introduction section.

“In a non-clinical pharmacokinetic study, the distribution of a test drug to various organs and tissues after a single or repeated dose and its change over time should be investigated. In the case of gas, unlike oral and injectable drugs, a non-clinical pharmacokinetic study with a single dose has not been performed. This was because there was no animal protocol for a single-dose study of the gas. The same is true for H2. The distribution of H2 to various organs and tissues after sustained inhalation has been studied in small animals such as rats. However, it was not possible to determine from these experiments whether H2 diffused from the lungs in a blood flow-independent manner or whether H2 was transported throughout the body in a blood flow-dependent manner. Therefore, in the present study, we devised an animal protocol for single-dose inhalation of gas and proved the latter to be true for the first time”.

(2) The treatment of pneumonia caused by COVID-19 has started in Wuhan, China. Please describe the expected pharmacological effects of hydrogen in this clinical setting.

The expected pharmacological effects of hydrogen in this clinical setting were described in the discussion section as follows.

“In patients with severe COVID-19, the immune over-response causes the production of large amounts of cytokines by alveolar macrophages, which becomes a cytokine storm, resulting in the progression of acute respiratory distress syndrome, abnormal blood coagulation, and multiple organ failure [21]. H2 gas not only inhibits the overproduction of cytokines [13], but also suppresses vascular endothelial damage [20], facilitates the flow of red blood cells in microvessels and increases the efficiency of gas exchange (M.S. unpublished observation). Accordingly, H2 inhalation therapy has great potential to improve the life expectancy of intubated COVID-19 patients admitted to the intensive care unit with severe hypoxemia. The HYBRID II Trial (Efficacy of inhaled HYdrogen on neurological outcome following BRain Ischemia During out-of-hospital cardiac arrest), a multicenter, randomized, double-blind, placebo-controlled, controlled clinical trial investigating the efficacy of H2 inhalation therapy for patients after out-of-hospital cardiac arrest, has been underway since 2017 using a hydrogenated oxygen supply device jointly developed by Keio University and Taiyo Nippon Sanso (jRCTs031180352)[4]. In the HYBRID II trial, patients after cardiopulmonary arrest and resuscitation have been treated with hydrogenated oxygen for 18 hours in combination with conventional cooling methods. Prior to Hybrid II, we had conducted an open-label, single-arm, prospective interventional trial at Keio University Hospital in Tokyo in 2014 to evaluate the feasibility and safety of H2 inhalation in patients with out-of-hospital cardiac arrest achieved a spontaneous return of circulation [5]. Non-cardiogenic cardiac arrest patients were also enrolled in this TRIAL; 5 patients were entered, one of whom was a CA patient due to severe pneumonia and septic shock. This patient had a stable respiratory state during H2 inhalation, but died after H2 inhalation was completed due to a rapid deterioration of the respiratory state. Based on this experience, we are considering a study protocol in which patients with severe COVID-19 disease will continue to receive hydrogenated oxygen for at least one week until their hypoxemia has sufficiently improved”.

---

## [Decision Letter · Decision Letter 1]

1 Jun 2020

Pharmacokinetics of a single inhalation of hydrogen gas in pigs

PONE-D-20-11712R1

Dear Dr. Sano,

We are pleased to inform you that your manuscript has been judged scientifically suitable for publication and will be formally accepted for publication once it complies with all outstanding technical requirements.

With kind regards,

Tohru Minamino, M.D., Ph.D.

Academic Editor

PLOS ONE

Additional Editor Comments (optional):

Reviewers' comments:

Reviewer's Responses to Questions

**Comments to the Author**

1. If the authors have adequately addressed your comments raised in a previous round of review and you feel that this manuscript is now acceptable for publication, you may indicate that here to bypass the “Comments to the Author” section, enter your conflict of interest statement in the “Confidential to Editor” section, and submit your "Accept" recommendation.

Reviewer #1: All comments have been addressed

Reviewer #2: (No Response)

Reviewer #3: All comments have been addressed

2. Is the manuscript technically sound, and do the data support the conclusions?

Reviewer #1: Yes

Reviewer #2: (No Response)

Reviewer #3: Yes

3. Has the statistical analysis been performed appropriately and rigorously? 

Reviewer #1: Yes

Reviewer #2: (No Response)

Reviewer #3: Yes

4. Have the authors made all data underlying the findings in their manuscript fully available?

Reviewer #1: Yes

Reviewer #2: (No Response)

Reviewer #3: Yes

5. Is the manuscript presented in an intelligible fashion and written in standard English?

Reviewer #1: Yes

Reviewer #2: (No Response)

Reviewer #3: Yes

6. Review Comments to the Author

Reviewer #1: The authors did a good job addressing the criticisms from the last review. This revised paper is now acceptable.

Reviewer #2: (No Response)

Reviewer #3: NO further comments. This paper is now acceptable.

This study is presented by a group that is leading the world in non-clinical and clinical

research on hydrogen gas.

7. PLOS authors have the option to publish the peer review history of their article (what does this mean?). If published, this will include your full peer review and any attached files.

Reviewer #1: No

Reviewer #2: No

Reviewer #3: No

---

## [Editor Report · Acceptance letter]

10 Jun 2020

PONE-D-20-11712R1 

Pharmacokinetics of a single inhalation of hydrogen gas in pigs 

Dear Dr. Sano:

I'm pleased to inform you that your manuscript has been deemed suitable for publication in PLOS ONE. Congratulations! Your manuscript is now with our production department. 

Kind regards, 

on behalf of

Professor Tohru Minamino 

Academic Editor

PLOS ONE